

# Repeatability of glucocorticoid hormones in vertebrates: a meta-analysis

Kelsey L. Schoenemann and Frances Bonier

Department of Biology, Queen's University, Kingston, Ontario, Canada

## ABSTRACT

We often expect that investigations of the patterns, causes, and consequences of among-individual variation in a trait of interest will reveal how selective pressures or ecological conditions influence that trait. However, many endocrine traits, such as concentrations of glucocorticoid (GC) hormones, exhibit adaptive plasticity and, therefore, do not necessarily respond to these pressures as predicted by among-individual phenotypic correlations. To improve our interpretations of among-individual variation in GC concentrations, we need more information about the repeatability of these traits within individuals. Many studies have already estimated the repeatability of baseline, stress-induced, and integrated GC measures, which provides an opportunity to use meta-analytic techniques to investigate (1) whether GC titers are generally repeatable across taxa, and (2) which biological or methodological factors may impact these estimates. From an intensive search of the literature, we collected 91 GC repeatability estimates from 47 studies. Overall, we found evidence that GC levels are repeatable, with mean repeatability estimates across studies ranging from 0.230 for baseline levels to 0.386 for stress-induced levels. We also noted several factors that predicted the magnitude of these estimates, including taxon, sampling season, and lab technique. Amphibians had significantly higher repeatability in baseline and stress-induced GCs than birds, mammals, reptiles, or bony fish. The repeatability of stress-induced GCs was higher when measured within, rather than across, life history stages. Finally, estimates of repeatability in stress-induced and integrated GC measures tended to be lower when GC concentrations were quantified using commercial kit assays rather than in-house assays. The extent to which among-individual variation in GCs may explain variation in organismal performance or fitness (and thereby inform our understanding of the ecological and evolutionary processes driving that variation) depends on whether measures of GC titers accurately reflect how individuals differ overall. Our findings suggest that while GC titers can reflect some degree of consistent differences among individuals, they frequently may not. We discuss how our findings contribute to interpretations of variation in GCs, and suggest routes for the design and analysis of future research.

Corresponding author
Kelsey L. Schoenemann,
kelsey.schoene@gmail.com

## INTRODUCTION

Since the development of immunoassays that allow the measurement of hormones in relatively small-volume tissue samples (*Ekins, 1960*; *Yalow & Berson, 1960*), the number of

studies investigating the patterns, causes, and consequences of endocrine trait variation has soared. Early work in this field described variation in hormone concentrations across species, populations, and life history stages (e.g., *Boswell, Woods & Kenagy, 1994*; *Klosterman, Murai & Siiteri, 1986*; *Pancak & Taylor, 1983*), while more recent work often measures among-individual variation in multiple endocrine traits, including hormone concentration, receptor density, binding protein concentration, and endocrine axis responsiveness (e.g., *Breuner et al., 2006*; *Bizon et al., 2001*; *Lattin & Romero, 2014*; *Liebl, Shimizu & Martin, 2013*). Thus, much of our understanding of how selection has shaped these traits derives from comparative studies that determine how conserved or variable hormones, receptors, or their effects are across taxa, or how those traits vary among individuals with geography, phylogeny, or other traits of interest (e.g., *Bókony et al., 2009*; *Eikenaar, Klinner & Stöwe, 2014*; *Heidinger, Nisbet & Ketterson, 2006*). Yet traits that exhibit adaptive plasticity, such as hormone titers, might not respond to selective pressures or ecological conditions as predicted by among-individual phenotype-fitness correlations (*Stinchcombe et al., 2002*; *Bonier et al., 2009*; *Bonier & Martin, 2016*). Moving beyond this comparative approach to better understand endocrine trait evolution requires knowledge about heritable individual differences in evolutionarily-important traits because natural selection acts upon this heritable variation at the individual level (*Bennett, 1987*; *Williams, 2008*). However, the extent to which variation in hormone levels can be attributed to fixed individual differences is poorly understood.

Concentrations of glucocorticoid (GC) hormones, for example, exhibit plasticity, here defined as the ability of a single genotype to produce multiple phenotypes in response to environmental changes, and referred to as flexibility in some contexts (*sensu Bonier & Martin, 2016*). The plasticity of GC titers helps organisms achieve allostasis (from the Greek *allo* meaning 'variable' + *stasis* meaning 'stand'; thus, 'stability through change'). Allostasis can refer to (1) the process that maintains or reestablishes the physiological parameters essential for life, such as pH, oxygen levels, or blood pressure (i.e., homeostasis), even while regulatory thresholds may change with environmental conditions, as well as (2) the organism's ability to produce the mediators, such as hormones, immune-signaling proteins, and (para)sympathetic activity, that promote physiological adaptation to new conditions (*Romero, Dickens & Cyr, 2009*; *McEwen & Wingfield, 2007*). In the case of GC hormones, rapidly elevating circulating concentrations (i.e., via activation of the hypothalamic-pituitary-adrenal [HPA] axis) promote behavioral and physiological changes that enable an organism to respond to and recover from acute challenges, and the modulation of baseline HPA activity supports responses to predictable changes in energetic demands across daily or seasonal cycles (*Sapolsky, Romero & Munck, 2000*; *Romero, 2004*; *Wingfield, 2005*; *Romero, Dickens & Cyr, 2009*). Failure to acknowledge, measure, or control for these sources of within-individual variation can diminish our ability to detect biologically significant patterns in GC secretion among individuals.

Estimating the repeatability (i.e., consistency over time or across contexts) of GC titers is one technique for assessing and potentially avoiding this pitfall. Multiple test statistics have been used to estimate the repeatability of a trait in a population (e.g., Spearman rank and Pearson correlation coefficients), but the intraclass correlation coefficient (ICC) is

the most prevalent in recent literature (*Sokal & Rohlf, 1995*; *Nakagawa & Schielzeth, 2010*). The repeatability of GCs within individuals can be used to determine the degree to which inferences made about GC measures may be generalized beyond providing information about the individuals at the time of sampling (e.g., *Bosson, Palme & Boonstra, 2009*; *Harris, Madliger & Love, 2016*; *Wada et al., 2008*). Moreover, repeatability itself may reflect the ability or strategy of an individual to cope with a challenge and, thus, is worthy of study in its own right (*Careau, Buttemer & Buchanan, 2014*; *Roche, Careau & Binning, 2016*). Finally, estimates of repeatability can approximate the upper limit of heritability of individual variation and, thereby, the extent to which natural selection can shape a trait (*Falconer & Mackay, 1996*; but see *Dohm, 2002*). Perhaps in recognition of these points, many studies have estimated the repeatability of GC measures (e.g., *Cook et al., 2012*; *Narayan, Cockrem & Hero, 2013a*; *Romero & Reed, 2008*; *Wada et al., 2008*). The availability of these estimates provides an opportunity to investigate whether GCs are generally repeatable across taxa, and how biological or methodological factors may impact these estimates.

To date, researchers have estimated the repeatability of GC levels in every class of vertebrates, and across various environmental contexts and spans of time. A meta-analysis of repeatability estimates across these studies could determine whether GCs are generally repeatable, and whether variation in the magnitude of repeatability can be explained by biological or methodological factors. For example, meta-analyses of behavior and metabolic rate repeatabilities have provided evidence of significant trait repeatability, as well as differences in repeatability according to sex, sampling interval, captive condition, and taxon (*Nespolo & Franco, 2007*; *Bell, Hankison & Laskowski, 2009*; *White, Schimpf & Cassey, 2013*). Here, we similarly seek to investigate sources of variation in estimates of repeatability of GCs. Specifically, the aim of this meta-analysis is to: (1) summarize the available evidence of repeatability of GC concentrations; and (2) identify biological and methodological factors that predict variation in the magnitude of GC repeatability.

## METHODS

### Literature search

We performed literature searches on Google Scholar between March 2016 and November 2017 using the terms: "repeatab\*," "consisten\*," "glucocorticoid," "cortisol", "corticoster\*", "repeated measure," and "individual variation." We identified 716 records in these searches. We screened the titles and abstracts of these records, looking for papers that estimated the repeatability (or 'consistency' or 'individuality') of concentrations of glucocorticoid hormones in a variety of tissues (e.g., blood, saliva, feces, feathers). To be selected for inclusion in this analysis, a study needed to have assessed repeated measurements from the same individual and estimated a repeatability coefficient (e.g., Spearman rank, Pearson, or ICC). We excluded duplicate and irrelevant articles and those that did not meet our inclusion criteria (Fig. 1). We also checked reference lists of selected papers to find additional studies that were not identified in the initial search. Lastly, we included three studies that collected repeated measurements of hormone concentrations from the same individuals but did not estimate repeatability, when we could obtain the original data to calculate repeatability.
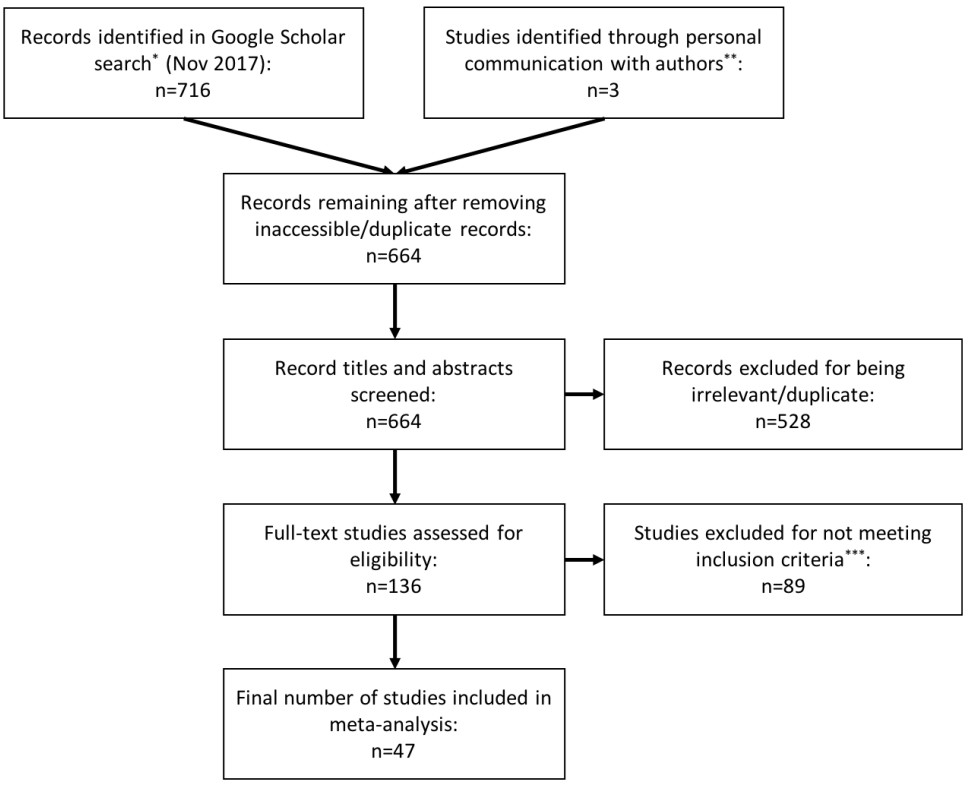

**Figure 1** **PRISMA flow diagram.** Preferred Reporting Items for Systematic Reviews and Meta-analysis (PRISMA) flowchart illustrating the process of study identification, screening, and inclusion in the meta-analysis. *We used the search terms: repeatab*, consisten*, glucocorticoid, cortisol, corticoster*, repeated measure, individual variation. **We included three studies that did not meet inclusion criteria (i.e., collected repeated within individuals, but did not estimate repeatability) because we were able obtain the original data from the study authors and calculate repeatability ourselves. ***We used the following inclusion criteria: the study had to assess repeated measurements of glucocorticoids within the same individual, and estimate a repeatability coefficient (e.g., Spearman rank, Pearson, or intraclass correlation coefficient).

## Repeatability estimates

We extracted repeatability estimates from the selected studies and categorized them as representing either *initial*, *response*, or *integrated* GC repeatability measures. We used the category *initial* to group repeatability estimates of GC titers measured in circulation within a time period expected not to reflect the acute stress of capture, *response* for repeatability estimates of the elevated GC titers following an acute capture, handling, or confinement stress, and *integrated* for repeatability estimates of GC titers that represent hormone secretion over a relatively long period of time (e.g., GC concentrations in feces, feathers, and saliva). If the study did not calculate repeatability, then, where possible, we obtained the original data and calculated an ICC repeatability, using the 'rptR' package (version: 0.9.2) in R (version 3.4.0, 2017-04-21) (*Nakagawa & Schielzeth, 2010*).

**Table 1** List describing how methodological and biological factors associated with each repeatability estimate were categorized for analysis.

| Factor | Categories |
|---|---|
| Time between measurements[a] | 0–7d, 8–14d, 15–30d, 31–90d, 91–195d, or 365+ |
| Number of measurements[a] | Two, more than 2 |
| Captive condition | Free-ranging, captive, wild-caught captive |
| Taxonomic class | Bird, mammal, amphibian, bony fish, reptile |
| Age | Adult, juvenile, both |
| Sex | Male, female, both |
| Life history stage (LHS) | Breeding, non-breeding, pre-breeding, NA[b] |
| Measured within LHS | Yes, No |
| Assay source | In-house, commercial kit |
| Assay tracer | Radioactive, enzymatic |
| Experimental manipulation[c] | Yes, No |
| Adjusted[d] | Yes, No |

**Notes.**

[a] Average, weighted by number of individuals when possible.

[b] We categorized life history stage as "NA" for domesticated or captive-born species because domestication can alter seasonal patterns in hormone physiology (*Donham, 1979*; *Sossinka, 1982*; *Künzl & Sachser, 1999*). Estimates from these species were not included in analyses that examined the effect of life history stage.

[c] Experimental manipulation refers to studies in which some or all individuals underwent a stressful manipulation intended to produce a response (not including routine capture and handling stress) at some point during the course of the study.

[d] Adjusted refers to whether or not estimates reflect GC repeatability after statistically controlling for factors expected to explain some of the variation in GC titers (e.g., year, sex, weather).

## Statistical analysis

We harvested information about several methodological and biological factors associated with each repeatability estimate and categorized these data for analysis (Table 1). We used linear mixed-effect models (LMMs) with the 'lme4' package (version: 1.1.13) to investigate variation in repeatability estimates. We included study identity as a random effect to control for potential bias arising from non-independence of multiple estimates derived from the same study (*Nakagawa & Santos, 2012*). One study, however, was coded with two independent study identities because the datasets included in this one study were collected by two different researchers, on different species, in different field sites (*Ouyang, Hau & Bonier, 2011*). We constructed separate LMMs to address each of the following questions with *initial*, *response*, or *integrated* GC repeatability measures:

1. *Does sampling regime predict repeatability?* To answer this question, we evaluated the following fixed effects: sample size, average time span between samples, and average number of samples.

2. *Does subject biology or sampling environment predict repeatability?* We evaluated the fixed effects taxonomic class, sex, whether samples were collected within or across life history stage, captive condition, and experimental manipulation (whether or not some/all individuals underwent a stressful manipulation intended to produce a response [not including routine capture and handling stress] at some point during the course of the study). We lacked sufficient power to evaluate the effect of age because we identified only two estimates of repeatability that were measured solely in juveniles or immature individuals. We also evaluated the fixed effect of life history stage (breeding,

non-breeding, or pre-breeding) in a subset of GC repeatability estimates measured within a single stage.

3. *Do laboratory or statistical techniques predict repeatability?* We evaluated the fixed effects use of an in-house assay or commercial assay kit, use of a radioactive or enzymatic tracer, and whether or not the statistical analysis incorporated confounding factors (i.e., if the repeatability estimate controlled for correlations between GCs and factors such as the time or year of sampling, and the breeding status, age, or body mass of the individuals sampled).

With the exception of models that included sample size as a fixed factor (Question 1, above), we weighted each estimate by its sample size to account for differences in statistical power among studies. Thus, estimates from larger studies had a greater influence in the models. We verified the normality of model residuals with a Shapiro test. When model residuals failed to meet the assumption of normality, we square-root transformed the data. To identify important predictors of repeatability, we coded global models with all candidate variables included as main effects and used the *dredge* function from the 'MuMIn' package (version: 1.15.6) to rank recombinant models with the Akaike's information criterion corrected for small sample sizes (AICc). We did not include any interaction terms in our models, due to small sample sizes. We report effect size and *p*-values from either the best-fit model or, when more than one model was ranked within 2 ΔAICc of the best-fit model, from a conditional average of all top models. Due to the small sample size of *integrated* measures available to address Question 2, we compared the saturated model to a null model using an *F*-test with Kenward–Roger approximation using the 'pbkrtest' package (version: 0.4-7) (*Kenward & Roger, 1997*; *Halekoh & Højsgaard, 2014*). For some non-ordinal variables (e.g., taxonomic class, sampling interval), it is more informative to consider the significance of the factor as a whole rather than at specific levels; therefore, in such cases, we performed a Type III ANOVA with Satterthwaite approximation for degrees of freedom using the 'lmerTest' package (version: 2.0-33) to obtain *p*-values (*Kuznetsova, Brockhoff & Christensen, 2016*).

In addition to including study identity as a random effect, we employed several other methods to address potential bias or pseudo-replication. First, we did not include redundant estimates from the same study nor re-analyses of the same data. Second, we assessed the independence of multiple repeatability estimates originating from the same study. If a single GC measure is correlated among multiple groups of individuals (e.g., similarly low *initial* GC repeatability in males and females from same population), then we might expect multiple repeatability estimates of the same population to be non-independent. To test for this effect, we performed a linear regression analysis with those studies that reported more than one estimate to test whether the number of estimates of repeatability in a study was associated with repeatability (*Nespolo & Franco, 2007*; *Bell, Hankison & Laskowski, 2009*). We did not find a relationship between *initial* repeatability and the number of estimates reported in the study (linear model: initial $n = 37$, $p = 0.127$), and no studies of *integrated* repeatability reported more than two estimates. We did find a significant negative relationship between the number of estimates of the repeatability of *response* GCs and their magnitude ($n = 31$, $\beta = -0.10$, $p = 0.002$), however, this relationship was driven

by a single study that reported multiple estimates of 0.00 repeatability. Thus, our inclusion of study identity as a random effect in all models was deemed sufficient to control for non-independence of multiple estimates from the same study.

Finally, to determine whether GCs are generally repeatable across all studies, we first needed to assess whether the estimates we obtained from the literature represent a random sample of the 'true' repeatability of GC titers. Given that the primary focus of most studies included in this analysis was not to estimate repeatability, we expect publication bias is unlikely to be an important source of bias for our results. Nevertheless, we assessed this and other potential biases directly by plotting every estimate against its sample size in funnel plots. Upon finding these plots symmetrical (Fig. S1), we concluded that bias is unlikely (*Egger et al., 1997*). Therefore, we calculated 95% confidence intervals around the mean repeatabilities of *initial*, *response*, and *integrated* measures across all studies, regardless of taxon, using 1,000 bootstrap samples of the data with replacement. We interpret a confidence interval that does not overlap zero as indicating that the mean GC repeatability estimate is greater than zero (i.e., the GC measure is, on average, somewhat repeatable), and interpret confidence intervals that do not overlap each other as indicating different repeatabilities.

## RESULTS

### Summary of the data set

We identified 47 studies that met our criteria for inclusion, from which we extracted 91 estimates of GC repeatability (summarized in Table 2, see Supplemental Information 2 for complete dataset). In brief, more estimates were made of *initial* or *response* measures than of *integrated* measures. The repeatability estimates included data from 36 species; however, more than two-thirds of the estimates originated from studies of birds. Free-ranging populations of adults with both sexes combined were more often studied than captive populations, juveniles or immatures, or separately for the sexes. About three-quarters of the estimates spanned a sampling interval of less than one year. The majority of estimates came from repeated measurement within the same life history stage and, of those measured within a stage, more were derived from measurements taken during the breeding season. Finally, the ICC was the most common repeatability estimate reported, with 42 studies reporting an ICC and only four reporting either Pearson or Spearman correlations; in one study, the authors did not clearly report method used nor respond to our requests for information.

### Repeatability of GCs

Overall, GC levels were moderately repeatable, with mean repeatabilities ranging from $0.230 \pm 0.041$ (SE) for *initial* measures, $0.320 \pm 0.058$ for *integrated* measures, and $0.386 \pm 0.041$ for *response* measures (Fig. 2). Moreover, the 95% confidence intervals around the mean repeatability of all three types of measures did not overlap zero (initial: 0.230 [0.162, 0.294], response: 0.386 [0.318, 0.449], integrated: 0.320 [0.235, 0.410]). As indicated by non-overlapping confidence intervals, the mean repeatability of *response* measures is greater than that of *initial* measures.

**Table 2 Summary of the data included in the meta-analysis.** Except for sample size, numbers provided reflect the number of estimates in each category.

| | | | | | | |
|---|---|---|---|---|---|---|
| GC measure | Initial[a] | Response[b] | Integrated[c] | | | |
| | 42 | 37 | 12 | | | |
| Sample size | Mean | Range | | | | |
| | 36 ± SE 4.5 | 8–352 | | | | |
| Sampling interval | 0–7d | 8–14d | 15–30d | 31–90d | 91–195d | 365+d |
| | 13 | 26 | 8 | 17 | 4 | 23 |
| Number of measurements | 2 | >2 | | | | |
| | 39 | 52 | | | | |
| Captive condition | Free-ranging | Captive-born | Wild-caught captive | | | |
| | 58 | 14 | 19 | | | |
| Taxonomic class | Bird | Mammal | Amphibian | Bony fish | Reptile | |
| | 60 | 11 | 8 | 9 | 3 | |
| Age | Adult | Juvenile | Both | | | |
| | 80 | 2 | 9 | | | |
| Sex | Male | Female | Both | | | |
| | 18 | 30 | 43 | | | |
| Life history stage (LHS)[d] | Breeding | Non-breeding | Pre-breeding | NA | | |
| | 36 | 21 | 9 | 25 | | |
| Within LHS | Y | N | NA[d] | | | |
| | 64 | 11 | 16 | | | |
| Assay source | In-house | Kit-based | | | | |
| | 51 | 35 | | | | |
| Assay tracer | Radioactive | Enzyme | | | | |
| | 42 | 44 | | | | |
| Experimental manipulation[e] | Y | N | | | | |
| | 21 | 70 | | | | |
| Adjusted[f] | Y | N | | | | |
| | 21 | 70 | | | | |

**Notes.**

[a] Initial GCs refer to concentrations of GCs expected not to reflect the acute stress of capture.

[b] Response GCs refer to elevated GC titers following an acute capture, handling, or confinement stress.

[c] Integrated GCs refer to GC titers representing hormone secretion over a relatively long time.

[d] We categorized life history stage as "NA" for domesticated or captive-born species because domestication can alter seasonal patterns in hormone physiology. Estimates from these species were not included in analyses that examined the effect of life history stage.

[e] Experimental manipulation refers to studies in which some or all individuals underwent a stressful manipulation intended to produce a response (not including routine capture and handling stress) at some point during the course of the study.

[f] Adjusted refers to whether or not estimates reflect GC repeatability after statistically controlling for factors expected to explain some of the variation in GC titers (e.g., year, sex, weather).

## Relationships between repeatability and biological or methodological factors

### Does sampling regime predict repeatability?

We found little evidence that sample size, time span between samples, or number of samples predicts GC repeatability. The null was the best-fit model for *integrated* measures and, while number of measurements and sample size were retained in top models of *initial* and *response* measures (Table S1), we did not find evidence that *initial* or *response* repeatability

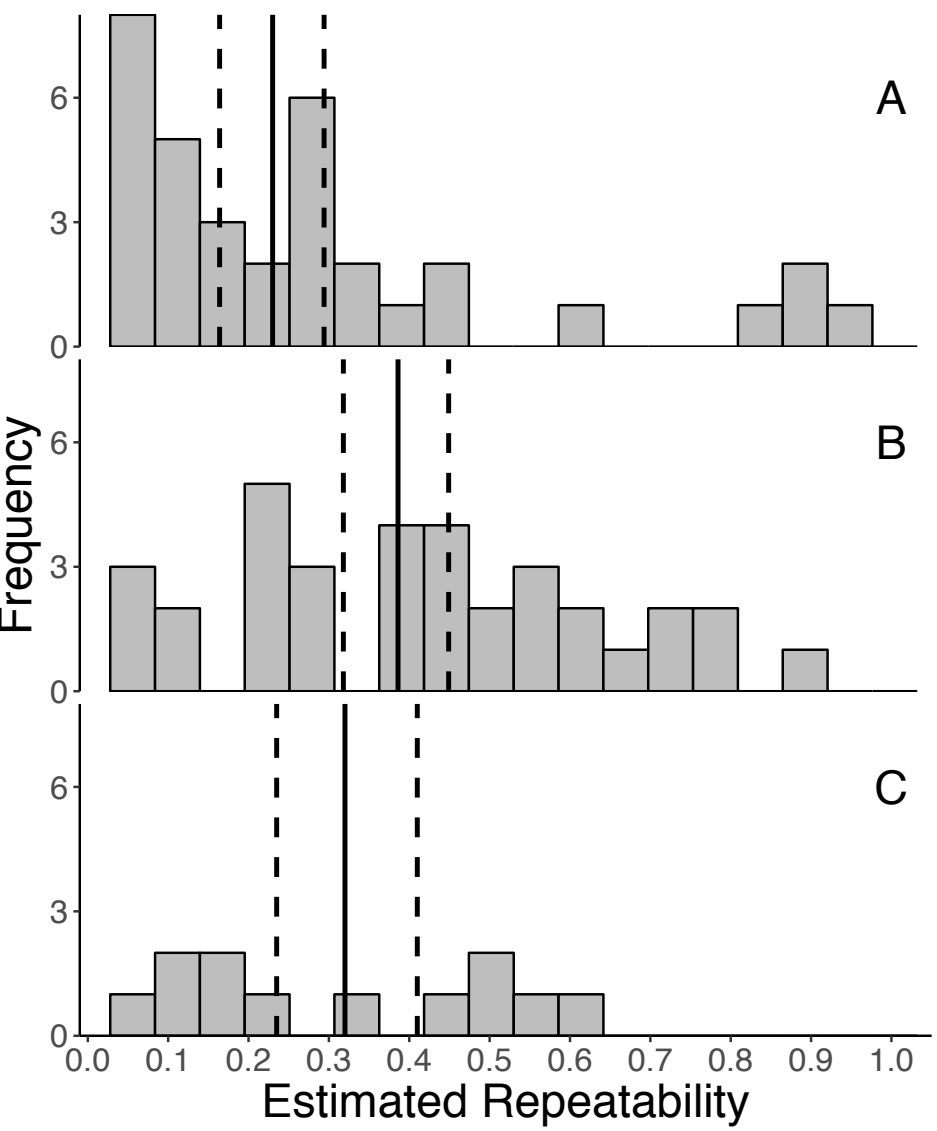

**Figure 2 Frequency distributions of all estimates of repeatabilities of (A) *initial*, (B) *response*, and (C) *integrated* glucocorticoid (GC) measures included in the meta-analyses.** The mean repeatability across all estimates of each category of GC is represented by a solid line, and the 95% CI (calculated from 1,000 bootstrap samples of the data with replacement) is represented by a dashed line. In this study, we defined *initial* measures as those representing GCs in circulation within a time period expected not to reflect the acute stress of capture, *response* for elevated GC titers following an acute capture stress, and *integrated* for GC titers that represent hormone secretion over a relatively long period of time (e.g., GC concentrations in feces, feathers, and saliva).

varied significantly with these factors (model average: all $p > 0.12$). Sampling interval, however, was retained in top models of *response* measures and, on average, repeatability was greater when repeated measurements were collected within 8–14 days of each other (0.607, $n = 8$), compared to either shorter (0–7 days; 0.327, $n = 5$) or longer (15–365 + days; 0.324, $n = 24$) intervals (Type III ANOVA; $n = 37$, $F(5, 35) = 2.840$, $p = 0.030$).

### *Does subject biology or sampling environment predict repeatability?*

Taxonomic class was retained in the top models explaining variation in repeatability estimates for both *initial* and *response* measures (Table S2). On average, amphibians had higher *initial* and *response* repeatability (0.833, $n = 4$; 0.786, $n = 4$, respectively) than birds (0.162, $n = 35$; 0.318, $n = 21$), mammals ([no *initial* GC repeatability estimates in mammals]; 0.446, $n = 5$), reptiles (0.270, $n = 1$; 0.21, $n = 2$), or fish (0.201, $n = 2$; 0.359, $n = 5$) (Fig. 3; Type III ANOVA; initial: $n = 38$, $F(3, 38) = 9.359$, $p < 0.0001$; response: $n = 27$, $F(4, 23) = 4.984$, $p = 0.005$). While sex was retained in the top models of *initial* measures, we did not find strong evidence that repeatabilities varied by sex (model average: all $p > 0.15$). Estimates of *response* repeatability were higher when derived from measurements within a life history stage (0.502, $n = 22$) than when derived from measurements across stages (0.072, $n = 5$) (Table S3; model average: $n = 27$, $\beta = 0.235$, $p = 0.007$). Neither experimental manipulation nor captive condition was retained in any top models. The global model evaluating *integrated* measures was not better-fit than the null ($F$-test: $n = 10$, $F(7, 3023) = 0.191$, $p = 0.988$).

Finally, in the subset analyses of repeatability estimates measured within a life history stage, we found little evidence that life history stage (breeding, non-breeding, or pre-breeding) predicts repeatability. The null model was the best-fit model for *initial* and *response* measures (Table S2). However, a univariate model including life history stage performed better than the null for *integrated* measures, where repeatability was on average higher in the non-breeding season (0.555, $n = 3$) compared to breeding (0.266, $n = 5$; $F$-test: $n = 8$, $F(1, 2370) = 10.7$, $p = 0.001$).

### *Do laboratory or statistical techniques predict repeatability?*

Assay type (in-house or kit) was retained in top models of *initial*, *response*, and *integrated* measures, while assay tracer was retained in the top models of *initial* and *integrated* measures (Table S4). Repeatabilities of *initial* and *integrated* hormone concentrations measured with RIA were lower than those measured with EIA, although this difference was not as evident for *initial* measures (Table S5; model average initial: $n = 40$, $\beta = -0.132$, $p = 0.071$; integrated: $n = 11$, $\beta = -0.194$, $p = 0.024$). In addition, the repeatabilities of *response* measures were lower when measured with a kit than those measured with an in-house assay, and tended to be lower for repeatability of *integrated* measures (Table S5; model average: response: $n = 35$, $\beta = -0.184$, $p = 0.040$; integrated: $n = 11$, $\beta = -0.172$, $p = 0.062$). Finally, whether or not confounding factors were controlled was retained in one top model of *response* measures, however, we did not find strong evidence that repeatability varied with this factor (Table S5; model average: $n = 35$, $\beta = 0.101$, $p = 0.340$).

## DISCUSSION

To better understand individual variation in GCs, we summarized published estimates of GC repeatability and identified factors that predicted the magnitude of those estimates. We found measures of *initial*, *response*, and *integrated* GCs had mean repeatabilities of 0.230, 0.386, and 0.320, respectively, with *response* repeatability estimates greater than *initial* repeatability. In general, this finding suggests that measures of GC titers reflect a

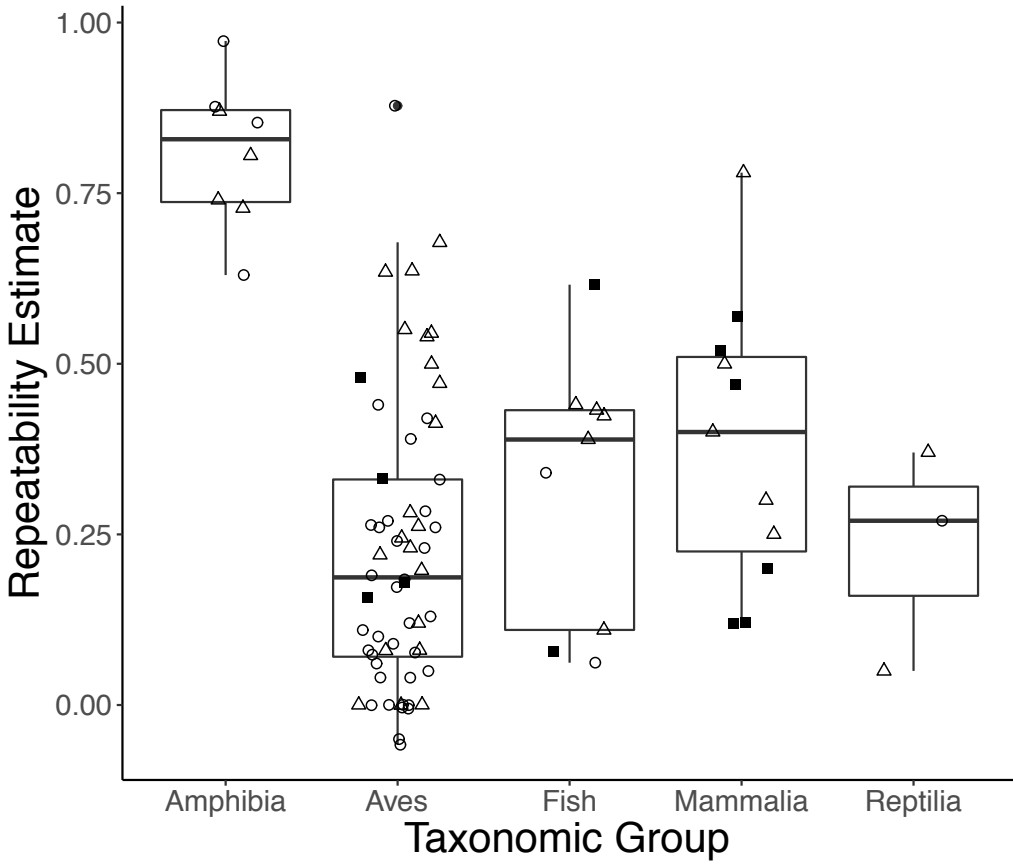

**Figure 3** **Boxplots showing variation in the average repeatability of all glucocorticoid (GC) measures across taxonomic classes (data are jittered along *x*-axis for ease of interpretation).** The plot's whiskers represent the 1.5 interquartile range, while the boxes represent the first and third quartiles, and the mid-line represents the median. Repeatability estimates for *initial* (open circles) and *response* (open triangles), but not *integrated* (closed squares), GC measures varied across taxonomic class (Type III ANOVA; initial: $n = 38$, $F(3, 38) = 9.359$, $p < 0.0001$; response: $n = 27$, $F(4, 23) = 4.984$, $p = 0.005$). In this study, we defined *initial* measures as those representing GCs in circulation within a time period expected not to reflect the acute stress of capture, *response* for elevated GC titers following an acute capture stress, and *integrated* for GC titers that represent hormone secretion over a relatively long period of time (e.g., GC concentrations in feces, feathers, and saliva).

moderate degree of consistent differences among individuals, however, some measures were more or less repeatable, depending on how the biological sample was collected and analyzed or which individuals were sampled. Specifically, we found that some estimates of GC repeatability were greater in amphibians, when all samples from an individual were collected within a single life history stage, and when samples collected within a life history stage came from the non-breeding season. We also found some evidence that GC repeatability was greater when hormone concentrations were measured using an in-house immunoassay, with an enzyme tracer, and when repeated measurements of the same individuals were collected across a relatively short time span (i.e., a sampling interval of 8–14 days).

The repeatability of GCs within individuals can be used to: (1) determine whether inferences made about GC measures may be generalized beyond the time of sampling (e.g., *Bosson, Palme & Boonstra, 2009*; *Harris, Madliger & Love, 2016*; *Wada et al., 2008*), (2) describe the ability or strategy of an individual to cope with a challenge (*Careau, Buttemer & Buchanan, 2014*; *Roche, Careau & Binning, 2016*), and (3) approximate the upper limit of heritability of individual variation and, thereby, the extent to which natural selection can shape a trait (*Falconer & Mackay, 1996*; but see *Dohm, 2002*). Below, we interpret our findings in light of each of these applications of estimates of repeatability.

While we found that some measures of GCs were highly repeatable (i.e., >0.70; see *Angelier et al., 2010*; *Ferrari et al., 2013*; *Narayan, Cockrem & Hero, 2013b*) and, therefore, expected to be reliable indicators of an individual's endocrine phenotype beyond the period of sampling, many other measures were not. Low repeatability may be caused by high within-individual variation, high measurement error, low among-individual variation, or a combination of all three. Whether a population exhibits low repeatability due to high within-individual variation (rather than low among-individual variation), or due to variation in trait consistency among individuals has different implications for how to collect and interpret data from that population of individuals (*Jenkins, 2011*; *Biro & Stamps, 2015*). When high within-individual variation is a concern, a single measurement of GCs will best capture individual differences when collected from all individuals instantaneously or while controlling for as many sources of environmental variation as possible. In the case of variation among individuals in trait consistency, a single measure of GCs will be unlikely to capture how individuals differ overall.

Whether or not an endocrine trait is repeatable for a given population, if individuals are sampled across different physical or social environments, or if some individuals differ in personality-related strategies, then the within-individual relationship between hormones and another variable of interest can differ from the population-level response in unexpected ways (*Roche, Careau & Binning, 2016*). For example, while a study found no relationship between brood size and baseline GCs among female tree swallows (*Tachycineta bicolor*), baseline GCs increased within individuals following an experimental increase in brood size (*Bonier, Moore & Robertson, 2011*). Additionally, olive flounder (*Paralichthys olivaceus*) with bold behavioral phenotypes responded physiologically to an acute stress in a manner opposite that of shy types, and these divergent responses were repeatable (*Rupia et al., 2016*). In both of these cases, failure to measure within-individual changes in GCs, or to recognize among-individual variation in the direction of those responses, would have obscured detection of the effects of the challenge of interest (i.e., brood size, acute stress) at the population level. Our finding of relatively low GC repeatability, particularly for *initial* GCs, strongly suggests that these measures frequently reflect an individual's short-term response to the environment more so than fixed differences among individuals.

Variation in GC repeatability can also be used to investigate differences in the ability or strategy of individuals or populations to respond to environmental change. For example, our finding of significantly greater repeatability in *response*, compared to *initial*, measures could indicate relatively greater canalization in the acute activation of the HPA axis, and a reduced plasticity of this trait within individuals. Consistent with this interpretation,

previous studies have estimated greater realized heritability of the GC response in genetic lines selected for high, rather than low, stress responses (*Brown & Nestor, 1973*; *Satterlee & Johnson, 1988*). Additionally, the greater repeatability of both *initial* and *response* GCs in amphibians could indicate different functions and/or responsiveness of the HPA axis in amphibians compared to other taxonomic classes (*Narayan, Cockrem & Hero, 2013a*). Finally, our finding greater repeatability of *response*, but not *initial*, GCs measured within a life history stage somewhat aligns with previous work, which has shown greater seasonal variation in baseline, rather than stress-induced, GC titers (*Romero, 2002*). And although our sample size was small ($n = 8$), our finding of greater repeatability of *integrated* GC measures during the non-breeding season seems to suggest less variation within individuals in the total secretion of GCs during that period, which could reflect a broader pattern of seasonal GC secretion across taxa.

If one aims to compare repeatability or trait consistency among individuals, populations, or even species, as described above, then an important consideration is whether variation among repeatability estimates is due to laboratory or statistical methodologies impacting within- or among-individual variation in the trait of interest. We found that some repeatability estimates were lower when measured with a commercial kit compared to an in-house assay, and when measured with an RIA as compared to an EIA. Commercial assay kits can be less precise (as well as less accurate) in measuring GC concentrations if they are not carefully validated for the study system (*Buchanan & Goldsmith, 2004*; *Sheriff et al., 2011*), which may explain lower repeatability estimates for GCs measured with kits. Further, the ease of use of commercial kits might lend itself to less precise lab practices than the more involved in-house assays. However, it is not clear why RIAs would be associated with lower repeatability. *Brown et al. (2010)* found that, while urinary cortisol assessed with either RIA or EIA exhibited qualitatively-similar temporal profiles, the RIA detected proportionally lower hormone concentrations (i.e., decreased among-individual variation). This lower among-individual variation could lead to lower repeatability, if it is not counteracted by simultaneously lower within-individual variation. Previous work has documented large inter-laboratory variation in measurements of absolute steroid hormone concentrations (*Bókony et al., 2009*; *Ganswindt et al., 2012*; *Feswick et al., 2014*; *Fanson et al., 2017*), suggesting that across-study comparisons of absolute values of individuals' GC titers are not valid. Finally, while we also found some evidence that *response* GC repeatability was greater when repeated measurements were collected over a relatively short time span (i.e., 8–14 days apart), even shorter time spans did not show a consistent pattern, and we did not detect a similar effect in any of the other GC measures. Overall, if one seeks to investigate the causes and consequences of variable GC repeatability among groups, to better understand the ability or strategy of these groups to respond to environmental conditions, methodological sources of variation must be considered and, ideally, controlled.

A final application of estimates of trait repeatability is to approximate the upper limit of heritability. The average repeatability of *initial* and *response* GCs reported here align well with the results of artificial selection and animal model approaches that estimate a similar degree of heritability in GC titers and the GC response (*Evans et al., 2006*; *Jenkins et al., 2014*; *Pottinger & Carrick, 1999*; *Touma et al., 2008*). These studies often find that

the heritability of baseline GCs is much lower than response GCs, if it is detectable at all (e.g., *Brown & Nestor, 1973*; *Satterlee & Johnson, 1988*; *Evans et al., 2006*). Thus, we expect baseline concentrations will be less likely to exhibit evolutionary change than stress-induced concentrations, when exposed to similar selective pressures. Furthermore, *Jenkins et al. (2014)* failed to find phenotypic or genetic correlations between baseline and stress-induced concentrations within individuals. This finding suggests that different mechanisms may control GC secretion during normal activity versus during challenging events, and that selection could affect variation in these traits independently (*Jenkins et al., 2014*). As a result, selective or ecological pressures should be expected to produce complex, context-dependent relationships between hormone titers and factors of interest. Altogether, the low-to-moderate repeatability and heritability of GC titers underscores the extent to which plasticity may generate individual variation, as well as the extent to which that variation may be transmitted to future generations.

While our meta-analysis of GC repeatability estimates allowed us to look for patterns in trait consistency across a range of methodological and biological factors, there are limitations to our dataset and thus our ability to draw strong inferences from it. For example, many studies calculated repeatability as a way to compliment or support their main results. If researchers are more likely to report repeatability estimates that support their main findings, then repeatability estimates available in the literature may overestimate true repeatability. In addition, our categorization of the biological and methodological data associated with each repeatability estimate could have over-simplified or otherwise misrepresented the reality of the study, which could make real patterns more difficult to detect, or possibly cause spurious patterns (e.g., among the more weakly-supported findings). Finally, sample size was limited for many categories included in our analyses, thereby reducing our statistical power to detect real patterns.

## CONCLUSION

Overall, this meta-analysis provides new insights into individual variation in GC titers, and highlights the importance of repeatability estimation to improve methods for collecting and interpreting biological data. We found that GCs were moderately repeatable, on average, but these estimates were also highly variable. Additionally, *initial* and *response* GC measures were more repeatable in amphibians than any other taxonomic class, while *response* GCs were more repeatable when measured within the same life history stage and *integrated* GC were more repeatable during the non-breeding season. We look forward to new research that further investigates how and why repeatability differs with these factors. However, our finding that laboratory techniques were also associated with variation in repeatability could serve as a reminder to be meticulous in monitoring for issues with the reproducibility of hormone data. Moving forward, a better understanding of endocrine trait evolution requires knowledge about heritable individual differences in evolutionarily-important traits. Our analysis shows that a single measure of individual variation in GC titers may not reflect how those individuals differ in general, and suggests different approaches to capture that signal, including repeated measurements of individuals both within and across environments.

## ACKNOWLEDGEMENTS

We thank Y Aharon-Rotman, A Gladbach, B Dantzer, J Riechert, C Vleck, H Wada, and S Winberg for providing data or additional information about their published studies included in the analyses. We also thank R Montgomerie for advice on the statistical analyses. We are grateful for the constructive remarks of N Cyr and two anonymous referees, which improved the final manuscript.

### Funding

Funding from a Natural Sciences and Engineering Research Council of Canada Discovery Grant (to Frances Bonier) and from Queen's University supported Kelsey L. Schoenemann's graduate stipend. There was no additional external funding received for this study. The funders had no role in study design, data collection and analysis, decision to publish, or preparation of the manuscript.

### Grant Disclosures

The following grant information was disclosed by the authors:
Natural Sciences and Engineering Research Council of Canada Discovery Grant.
Queen's University.

### Competing Interests

The authors declare there are no competing interests.

### Author Contributions

- Kelsey L. Schoenemann conceived and designed the experiments, performed the experiments, analyzed the data, wrote the paper, prepared figures and/or tables, reviewed drafts of the paper, compiled dataset.
- Frances Bonier conceived and designed the experiments, contributed reagents/materials/analysis tools, wrote the paper, prepared figures and/or tables, reviewed drafts of the paper.

### Data Availability

The raw data is provided in a Supplemental File.

### Supplemental Information

Supplemental information for this article can be found online at http://dx.doi.org/10.7717/peerj.4398#supplemental-information.

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
