# Peer review of "Repeatability of glucocorticoid hormones in vertebrates: a meta-analysis"

_PeerJ, doi:10.7717/peerj.4398_

## Round 0.1 · original submission · Minor Revisions

We have received three reviews for your manuscript and all of them were quite positive.

All reviewers agreed that the manuscript was well written, that results were well presented and limitations were recognised. Yet, the reviewers also provided a few relevant minors comments that should be relatively easy to integrate when revising your manuscript.

Overall, this manuscript will make a good contribution to the field.

·

Basic reporting

Summary - The purpose of this paper was to analyze the available literature regarding the repeatability of glucocorticoid measurements. The study included measurements of baseline (initial), acute stress-induced (response), and integrated glucocorticoids. The analysis was conducted on different taxa with birds being the best represented by number of studies. Findings were that initial glucocorticoid measurements were moderately repeatable with response and integrated measure being more repeatable than initial measures. Factors contributing to the repeatability of glucocorticoid measures included whether the measurements were taken within a life history stage and what type of assay was used with a kit producing the least repeatable results. Of the taxa analyzed, amphibian measures of initial and response glucocorticoids were the most repeatable.

The writing is clear and flows well.

The Introduction provides necessary background.
Minor comment – it would be nice to define allostasis for readers whose expertise fall outside of this field.

The literature was well used and cited.

Structure is within PeerJ standards.

Figures are well done.

Raw data was provided.
Minor comment – it would be nice to see the r2 values for the analyses of Supplemental Table 1. even if only for the best fit models (those within 2 ΔAICc).

Experimental design

The experimental design was a Meta-analysis of the available literature on repeatability of glucocorticoid measurements. I think that the study was well designed. Explanation of statistical analyses was clear.

Validity of the findings

Sample sizes for some of the analyses were small. For example, amphibians showed substantially greater repeatability estimates compared to the other taxa, but their numbers were small (n=4). In general, I think that small sample sizes limit this study. Having said that, I think that the study is well done and provides interesting results with the data that they do have. I am also very surprised that RIAs were less reliable than ELISAs in terms of repeatability in glucocorticoid measurements. However, the authors address this in the discussion.

Additional comments

As stated above, small sample sizes hindered this study for certain analyses. However, the study was well done, and it is an important topic to the field. Overall, I enjoyed reading this paper and I recommend it for publication.

It does, however, leave me with questions. For example:

1) For the integrated glucocorticoid analyses: Were there differences between the repeatability of glucocorticoids taken from feathers vs. feces (or saliva)?

2) The authors explored the influence of pre-breeding, breeding, and nonbreeding on the repeatability of glucocorticoids, but it would be nice to see what is happening within these time periods (e.g. for birds, within breeding – egg laying, incubating, nestling provisioning).

It would be great to be able to address these questions if sample sizes permit.

Reviewer 2 ·

Basic reporting

No Comment

Experimental design

No Comment

Validity of the findings

No Comment

Additional comments

This study is a meta-analysis investigating the repeatability of corticosterone across taxonomic groups. The authors find varying degrees of repeatability across taxonomic groups, assay type, sampling interval.
General comments.
Overall I think this is a well written manuscript and the authors acknowledge the limitations of this type of study. I only have very general comments which are more of me thinking about the limitations of this type of study.
I think this is a question that is worth addressing. My worry with these types of studies, or even the individual studies that were used, may often underestimate repeatability because differences in environmental conditions, stage of breeding, or assay variation. The authors of the current submission attempt to control for some of these factors but did the authors in the main studies? I think that some species lend themselves very nicely to these types of studies which I would venture to guess they show a smaller range in of glucocorticoid concentrations across the annual cycle. For instance, white-crowned sparrows show high seasonal variation in glucocorticoid titers so unless you capture the bird at the exact same stage of its breeding cycle you likely will not find that corticosterone is repeatable. But this may be a classic example in which we fail to reject the null hypothesis when it is not true simple because our sampling date was not correct to capture the repeatability of an endocrine measure. Alternatively, it could be that is just not that repeatable because environmental conditions are so variable that not matter what we do we will find relatively low repeatability. This then makes me think that there are likely less studies on repeatability then we would like because the limitations of the data are known and cannot be properly addressed. The authors have acknowledged many of these caveats in the discussion so I must commend them for that.
I understand why Wingfield and McEwen have pushed the concept of allostasis from a point of understanding how corticosterone levels will fluctuate across life history stage, environmental condition, and etc. In terms of the repeatability measure one would assume that corticosterone levels both at baseline or initial sample and in response to a stressor would fluctuate greatly. There is very likely a range of reactions as nicely outlined by Taft and Vitousek 2016 or Angelier and Wingfield 2013 so if all conditions are similar enough across our sampling points then we would detect a high degree of repeatability (potentially).
It would be nice to see a meta analyses of repeatability in which individuals are challenged. This would get at what factors can decrease the repeatability of glucocortiod measures. Some of the studies that may really highlight these changes are food restriction/removal studies, environmental extremes such as storms, selection for stress physiology across a breeding range, light pollution, increases in brood size and etc. The combination of these factors may begin to shed light on factors in the environment that drive changes in stress physiology. These would all provide examples were metabolic demands have led to changes in corticosterone concentrations. To put it into the authors words, we could look at the phenotypic plasticity of hormonal signaling or reaction norms. This obviously is not something I would expect to be add to this paper but rather expanding on in a future paper. I think this would bring some valuable insight into the concept of allostasis in terms of using a meta-analysis to find common reactions to environmental challenges.
I think the authors should have a look at and cite the following paper as it is important for thinking about corticosterone assays across labs. Although this particular meta analyses really controls for laboratory biases. IT would be relevant for the paragraph starting on line 343. Fanson et al. 2016 Inter-laboratory variation in corticosterone measurement: Implications for comparative ecological and evolutionary studies

Reviewer 3 ·

Basic reporting

This manuscript reports results from a well designed study using data from the literature. The manuscript is well written and the results are well presented. The study will be a useful addition to the literature on repeatabilities of glucocorticoid measurements in animals.

Experimental design

The study used a wide range of statistical methods. Whilst the reviewer is not able to comment on the validity of these methods, the appropriateness of calculating mean repeatabilities after 1000 bootstrap samples (line 196) is questioned. The authors should include mean and SE values calculated directly from the repeatabilities. These mean and SE values should be reported as well as or instead of the bootstrap values. It is misleading to report in the abstract that the bootstrap values were mean values.

Validity of the findings

A comprehensive dataset has been collated from the literature. The conclusions are supported by the evidence.

---

## Round 0.2 · accepted · Accept

Thank you for performing the revisions required to your manuscript. It is now acceptable for publication in PeerJ and provides a nice contribution to the field.